# Assessment of Quality in Antimicrobial Calcium Phosphate Research (AQUACAP): A Systematic Review

**DOI:** 10.3390/ma18071543

**Published:** 2025-03-28

**Authors:** Robert Kamphof, Jacobus Arts, Giuseppe Cama, Rob Nelissen, Bart Pijls

**Affiliations:** 1Department of Orthopaedics, Leiden University Medical Center, Albinusdreef 2, 2333 ZA Leiden, The Netherlands; r.g.h.h.nelissen@lumc.nl (R.N.); b.g.c.w.pijls@lumc.nl (B.P.); 2CAM Bioceramics B.V., Zernikedreef 6, 2333 CL Leiden, The Netherlands; giuseppe.cama@cambioceramics.com; 3Department of Orthopaedic Surgery, Maastricht University Medical Center, P. Debyelaan 25, 6229 HX Maastricht, The Netherlands; j.arts@mumc.nl; 4Department of Biomedical Engineering, Orthopaedic Biomechanics, Technical University Eindhoven, De Rondom 2, 5600 MB Eindhoven, The Netherlands

**Keywords:** hydroxyapatite, coating, infection, prosthetic joint, checklist

## Abstract

The effectiveness of antimicrobial ion-substituted calcium phosphate biomaterials has been investigated in numerous studies, but reporting guidelines and quality checklists are missing. A novel quality checklist was created for assessing reporting and methodological quality by experts of relevant disciplines. The checklist consisted of 20 items for reporting quality (maximum score 32) and 11 for methodological quality (maximum score 18). The checklist was subsequently implemented to assess the reporting and methodological quality of 58 studies in this field. Possible associations between study quality, year of publication and citations were investigated, and items for improvement were identified. Main items for improvement in reporting quality (average score 18/32) were reporting variability and statistics of data, reporting rationales for study design and the clinical relevance of the outcomes. Methodological quality (average score 11/18) could be improved by including positive control groups, using clinically relevant material formulations and including tests of the material toxicity. No association was found between study quality and year of publication. Methodological quality was associated with a higher number of annual citations. This study identifies key areas for improvement of reporting and methodological quality in the field of ion-substituted antimicrobial calcium phosphates. With these findings, the quality of future studies on antimicrobial CaP materials can be improved. The new quality checklist can also be used to improve study design for future research and enables better comparison between study outcomes.

## 1. Introduction

A growing number of patients worldwide depend on implantable medical devices, such as hip and knee replacement implants, to maintain or enhance their quality of life. A major complication after prosthesis implantation is the occurrence of implant-related infection, which has a huge impact on patients’ quality of life [1,2,3]. Furthermore, the odds of adverse outcomes after prosthesis-related infection are increasing due to the rise of antimicrobial resistance to antibiotics (AMR) [4,5]. Consequently, there is a rich field of studies dedicated to seeking new implantable biomaterials that can prevent implant infection. One of the materials under investigation is calcium phosphates (CaPs). CaPs are a class of ceramic biomaterials characterised by their chemical similarity to natural bone [6]. This similarity gives CaPs excellent biocompatibility and allows them to be resorbed and remodelled to regenerate bone tissue [6,7]. CaPs have a variety of clinical uses, such as implant coatings or bone void fillers. By substituting foreign elements into these materials, the biological properties of CaPs can be enhanced in a variety of ways [8,9]. Notably, the addition of silver, copper and zinc ions into the CaP structure has been shown to imbue these materials with antimicrobial properties [10,11]. As CaP materials are already employed in clinics, a CaP product with antimicrobial properties could be adopted rapidly by surgeons. However, while there are numerous in vitro studies on this subject, the is currently no dedicated tool to assess the study quality [12].

Study quality encompasses two aspects: quality in reporting and quality in methodology [13]. Low study quality, caused by a lack of reporting or methodological quality (or both), results in research waste and/or the publication of misleading or biased results [14]. Reporting quality refers to the readability, replicability and completeness of a study. Methodological quality refers to the reliability of study outcomes, based on its study design characteristics [15]. High methodological quality means that the experiments are properly designed and executed, resulting in reliable results [14]. In biomaterial design, this means adequate physiochemical characterisation, unambiguous testing of the biological properties (antimicrobial effect and cytotoxicity), use of proper control groups (test verification) and replicate testing (test validation). Quality in reporting concerns the reproducibility and accessibility of the results as well as reporting on the rationales for different elements of study design [16]. Sufficient reporting quality is necessary to interpret the reported results and assess their scientific impact [16]. Moreover, adequate reporting quality is required to accurately assess the methodological quality. Conversely, a high methodological quality is not necessary to judge the reporting quality. Both reporting and methodological quality influence the study validity when comparing or combining outcomes across studies.

Methodological variability hampers uniform assessment of the antimicrobial effect of ion-substituted CaPs across studies. This heterogeneity between studies was the main challenge of an earlier systematic review on antimicrobial CaPs [12]. Methodological variability can be caused by a lack of reporting guidelines or quality checklists on this topic. Quality checklists are essential tools to ensure uniformity in methodological practices and to help eliminate bias [17]. While various tools already exist to assess the quality or risk of bias of clinical and preclinical studies, currently no tools are suitable for material design studies with in vitro data [18]. Therefore, this study has two goals:To develop a novel checklist for assessing essential quality items of preclinical studies on antimicrobial ion-substituted CaPs by a panel of subject matter experts.Using this checklist to assess the quality of existing literature and identify areas for improvement.

## 2. Materials and Methods

This systematic review was conducted in line with the PRISMA guidelines for systematic reviews [19]. No protocol was registered for this review, since it represents a continuation of earlier research, which was published on the 26 May 2023 [12]. The protocol for our earlier review was registered on the Harvard Dataverse on the 6 November 2021, at https://doi.org/10.7910/DVN/HEP18U.

A structured approach was used to assess the study quality of antimicrobial CaP research. First, a quality checklist was created by a panel of experts, which can be used to evaluate scientific literature in terms of reporting and methodological quality. Then, a set of 58 papers was scored using the checklist [20,21,22,23,24,25,26,27,28,29,30,31,32,33,34,35,36,37,38,39,40,41,42,43,44,45,46,47,48,49,50,51,52,53,54,55,56,57,58,59,60,61,62,63,64,65,66,67,68,69,70,71,72,73,74,75,76,77]. Finally, statistical analysis was used to describe the results and find trends within the data. Further analysis was used to check for possible associations between study quality and publication year or between study quality and number of citations.

### 2.1. Design of the Checklist

As a basis for the checklist, the OHAT Risk of Bias Rating Tool for Human and Animal Studies was used [78]. The OHAT tool is used to assess the internal validity of in vivo studies and consists of 11 questions regarding the characteristics of the study groups, exposure to the administered drug or treatment, assessment and reporting of the outcome and statistical methods used.

Experts from relevant fields (orthopaedics, infectious disease, biomaterials and the medical device industry) were invited to provide input on the most important aspects of reporting and methodology in this field during several online sessions. All experts are listed as co-authors in this publication or mentioned in the acknowledgements section. During these sessions, the minimum quality standards for papers reporting on antimicrobial CaP materials was established from academic, manufacturing and clinical perspectives. A list of quality items (based on OHAT) was identified that were relevant for in vitro testing of antimicrobial ion-substituted CaP materials. The quality items were formulated into multiple-choice questions and stratified as reporting items or methodological items for the final checklist. All quality items were subdivided into the following topics for both reporting and methodological quality:Use and reporting of positive and negative control groups.Material characterisation, especially phase characterisation and elemental analysis.Reproducibility and validity of the antimicrobial tests.Reporting and discussion of the results of antimicrobial tests.Use of replicate experiments and statistics.Testing of material toxicity.Rationales for experiment design choices.

A scoring system for the checklist was based on AQUILA (Assessment of QUality In Lower limb Arthroplasty), a checklist for assessing the reporting and methodological quality in total joint arthroplasty studies [13]. The purpose of the total score was to provide a quality mark for in vitro studies on antimicrobial CaPs. For each quality aspect, a number of possible multiple-choice answers were formulated, and scores were assigned to each answer. Answers representing insufficient study quality scored 0 points, and sufficient quality scored 1 point. When more detailed analysis or more complex tests were provided than the expected minimum, 2 points were awarded. For example, in reporting quality, reporting the number of replicate experiments is scored with 1 point, while failing to do so results in 0 points. If the reporting is clear enough that the reader can distinguish between biological and technical replicates, the study scores 2 points. As a second example, in methodological quality, a minimum of 3 replicates is needed to score 1 point, while having more than 9 replicates is awarded with 2 points.

### 2.2. Implementation of the Checklist

Fifty-eight studies were randomly selected from the dataset of our earlier review (108 studies) that assessed the antimicrobial effect of ion-substituted CaPs [12]. The original dataset was selected from a literature search containing 1016 hits according to a set of exclusion and inclusion criteria, reported in more detail in the original study. In brief, exclusion criteria were as follows: if the materials were not ion-substituted, if no quantitative measure of antimicrobial effect was reported or if any material of chemical phase other than calcium phosphate was present. Studies were also excluded if the full text could not be retrieved, if they were written in a language not read by the authors, if the data were unintelligible or if the number of repeat experiments was unclear (studies where no error bars/variance were reported were assumed to have N = 1 and were included). The PRISMA flow chart for this dataset can be found in the original publication [12]. From the original 108 studies, 8 were used to optimise the quality checklist. Of the remaining 100 studies, 50 studies were randomly selected to give a total of 58 included studies. These 58 studies were assessed for their reporting quality and methodological quality.

Each study was reviewed according to the checklist. In case multiple experiments were performed, the highest applicable score for all results was chosen. In questions on control groups, ‘negative control’ was defined as a group that did not have any antimicrobial effect, such as CaP material without antimicrobial ions, an empty culture vessel or an uncoated piece of metal. ‘Positive control’ was defined as a group that was certain to have an antimicrobial effect, such as antibiotics. When there were multiple valid negative controls (e.g., both an empty group and a non-substituted CaP group), non-substituted CaP was considered the negative control. The database recording the quality items for included literature was created using IBM SPSS version 29.

Total scores for reporting and methodological quality were calculated using Microsoft Excel (version 2308). The database and score lists used to produce the data for this review were posted on the Harvard Dataverse on 5 December 2024, at https://doi.org/10.7910/DVN/KTD9EG. Studies were divided into quartiles based on their reporting quality scores to identify high- and low-scoring studies. The results for the upper (Q1) and lower quartile (Q4) were compared to identify areas where reporting quality was consistently high or low and where it could be improved.

### 2.3. Statistical Analysis

Standard descriptive statistics were used for the reporting quality and methodological quality. Linear regression was used to assess the presence or absence of association between reporting and methodological quality scores.

Linear regression was used to investigate trends in study quality over time (study age: baseline year 2024 minus publication year). Association between citation rate and study quality was also investigated using linear regression. The citation rate was defined as the total number of citations (Google Scholar, obtained 3 July 2024) divided by the study age. The citation rate was used instead of absolute citations to avoid confounding with trends based on study age. The obtained slopes and 95% confidence intervals were reported. Linear regressions and creation of graphs were performed using the R programming language (version 4.2.0) in RStudio (version 2022.02.2 + 485).

## 3. Results

### 3.1. Design of the Checklist

The experts identified a total of 31 relevant items: 20 items were considered important for reporting quality, and 11 were considered important for methodological quality (see Table 1 and Table 2). The maximum score for the reporting quality was 32, and is was 18 for methodological quality. The full checklist with possible answers and corresponding scores was published on the Harvard Dataverse on 10 December 2024, at https://doi.org/10.7910/DVN/TYGLAU.

### 3.2. Implementation of the Checklist

The reporting quality of 58 studies was assessed. The results per reporting quality item are shown in Table 3. The scores for reporting quality ranged between 12 and 23 (max 32, average 18, Q4: 12–16, Q1: 20–23). The database containing data on the scores of all 58 included studies was published to the Harvard Dataverse on 5 December 2024, at https://doi.org/10.7910/DVN/KTD9EG.

Papers with reporting quality scores in the highest quartile, Q1, scored consistently higher for nearly all reporting quality items than papers with the lowest reporting quality scores, Q4. The largest difference between Q4 and Q1 was in the reporting of antimicrobial testing conditions. Fourteen studies from Q1 reported the test conditions in a reproducible way, versus only 5 studies in Q4. Other prominent differences between the upper and lower quartile were as follows:The reporting of the starting number of microorganisms;Reporting the antimicrobial test results on a logarithmic scale;Reporting a rationale for the chosen microorganisms.

Both Q4 and Q1 scored well on these quality items:Reporting of negative control groups;Material characterisation using X-ray diffraction (XRD);Material characterisation using other methods;Reporting the used microbial species.

Conversely, consistent low scores were obtained for these quality items by most studies:Reporting rationales for the design of antimicrobial tests;Reporting a measure of variance for the antimicrobial test results;Discussing the clinical relevance of the reported results.

Results for each methodological quality item are shown in Table 4. The methodology scores ranged between 5 and 16 (max 18, average 11, Q4: 5–9, Q1: 12–16). In methodological quality, the upper quartile (Q1) scored better than the lowest quartile (Q4) only for certain specific quality items. This is in contrast to reporting quality, which was consistently higher across all quality items. Specifically, Q4 scored higher in four quality items:
Testing the elemental composition of the material;Testing for ion release;Performing more replicate experiments;Testing for material toxicity.

Notably, these four methodological quality items all involved performing additional tests.

Methodological quality was consistently high regarding material characterisation by XRD and other methods. In contrast, studies from both Q1 and Q4 scored low on the use of positive control groups (one study in Q4 and two in Q1). While most studies performed antimicrobial experiments on both Gram-positive and Gram-negative bacteria (36 studies), a much smaller group of studies included fungal microbes (11 studies). Furthermore, only five studies enumerated both adherent and planktonic bacteria, while most studies only enumerated one of the two.

### 3.3. Statistical Analysis

The total combined scores for reporting and methodological quality ranged between 19 and 38 (max 50, average 29, Q4: 19–26, Q1: 31–38). No relevant association was found between reporting and methodological quality scores (see Figure 1).

Publication year ranged from 2007 to 2021, and publication age ranged from 3 to 17 years. The number of citations ranged from 1 to 308, and the citation rate varied from 0.25 to 22.

As expected, an association was found between study age and total citations (slope 8.12, 95% CI 5.10–11.14, see Figure 2A). This association was not observed between study age and the citation rate (slope 0.19, 95% CI −0.16–0.54, see Figure 2B). To avoid confounding between absolute citations and study age, the citation rate was used for further analysis.

No relevant association was found between reporting quality and study age or citation rate (see Figure 3A,C and Table 5). Likewise, study age was not associated with methodological quality (see Figure 3B). However, there was an increase of 0.14 annual citations per point increase in methodological quality (see Figure 3D). This means that studies with a higher methodological quality are expected to receive more citations per year on average.

## 4. Discussion

A checklist for assessing the reporting and methodological quality of studies reporting on antimicrobial performance of ion-substituted CaP materials was developed. This checklist can be used to assess the quality of existing studies and guide future research. For instance, the identified methodological items may help researchers choose more methodological sound methods in the design of new studies. The reporting quality checklist may be used to check if all relevant items are reported, similar to, e.g., the CONSORT or STROBE checklists, and may be submitted as an appendix for future studies on CaPs.

Using the checklist to score 58 studies, the reporting and methodological quality of the research field was assessed. There was a large variety in the scores obtained, and no association was found between the reporting quality scores and methodological quality scores. This indicated that the checklist can independently assess both methodological and reporting quality. Additionally, high methodological quality does not necessarily lead to high reporting quality or vice versa.

### 4.1. Areas of Improvement in Reporting Quality

No studies achieved exceptionally high or low total scores for reporting quality. In terms of individual quality items, many studies scored high for reporting control groups and material characterisation. However, there are three areas most in need of improvement.

Firstly, reporting on the clarity and reproducibility of the experiments must be improved. A sizeable number of studies failed to report essential study design characteristics, such as the starting number of microorganisms or other experimental conditions. Moreover, there were mismatches between the ‘methods’ and ‘results’ sections in some studies. In some cases, results were shown without (adequate) description of the experiments in the methods, or the experiments were described in the methods, but the corresponding results were not shown. It is essential that future studies improve reporting on these items, since the reproducibility of the experiment depends on this information. Fortunately, these aspects of reporting quality should be relatively straightforward to improve.

Secondly, the rationales for all aspects of study design are often missing. It was often unclear why the antimicrobial CaP materials were tested in their present form (usually as powders, which do not have a direct clinical application), why they were challenged against the chosen microorganisms or why a specific type of test was chosen. The reason for this lack of rationales is unknown, but it can be caused by choosing the wrong research question or not having a research question at all [14]. Research methodology based on scientifically accepted rationales can be based on standards published by the International Organisation for Standardisation (ISO) or the American Society for Testing and Materials (ASTM) [79,80,81]. Aside from improving reporting quality, clear rationales can also improve methodological quality by guiding the design of experiments in an early stage.

Thirdly, and related to the second issue, more critical discussion of the results is needed. Several included studies claimed ‘impressive’ or ‘significant’ outcomes but failed to define these terms. Consequently, the clinical relevance of these results is unclear, leading to research waste [14]. One potential root cause of this issue is a lack of clinical outlook in these studies. It is important to interpret results from CaPs in a clinical context or clinical scenarios in order to advance the clinical application of antimicrobial CaPs. In this regard, it is critical to realise that statistical significance is not the same as clinical significance [82,83]. Statistical significance is an indicator of the likelihood that study outcomes rely on chance, but it does not automatically mean that these outcomes are interesting or useful. Ideally, a clinically or microbiologically relevant target outcome should be formulated at the onset of the study [83]. This target outcome can be compared to the measured outcomes in order to determine their clinical/microbiological relevancy. The target outcome may vary from study to study, based on material formulation, methodology and research question. As an example of adequate reporting of this quality item, we would like to commend the study by O’ Sullivan et al. for its excellent discussion of the experimental outcomes [26]. There, it is stated that an antimicrobial material should show at least a 3-log reduction in CFU count in vitro to be considered clinically relevant, although total eradication is preferred. Because of the stated target outcome, it is easy to verify if the study achieved its aims and to compare it to other studies with the same goal. However, it is important that this target relates to a clinically relevant situation or desired clinical outcome, even though it has been shown that this is not always straightforward [84].

### 4.2. Areas of Improvement in Methodological Quality

Most studies obtained high scores on physiochemical characterisation. In nearly all studies, phase characterisation by powder X-ray diffraction (XRD) was performed. XRD provides information on phase purity and crystallinity and, as such, is one of the most important characterisation techniques in this field. Because of its significance in the field, XRD was identified as an important quality aspect separate from other physiochemical characterisation methods. Aside from XRD analysis, other characterisation methods were also performed, albeit less frequently. The most common secondary characterisation technique was Fourier-transform infrared spectroscopy (FTIR). FTIR allows for the detection of various impurities in the synthesised CaPs. Although there was much variability in the characterisation techniques performed, most studies performed at least one other technique aside from XRD. In contrast, there are three aspects of methodological quality where quality needs to be improved.

Firstly, only a few studies included positive control groups in antimicrobial performance assays. A potential reason for the exclusion of positive controls is that they do not appear to contribute directly towards answering the research question. However, positive controls are helpful to assess the validity of the experiment, because they confirm that the experiment is able to generate a favourable outcome. Since positive controls do not add significant workload to an experiment, future research should be able to include positive controls.

Secondly, more physiochemical tests are necessary to confirm the composition of the ion-substituted CaPs, such as elemental composition and ion release experiments using relevant elemental analysis techniques like ICP-MS, ICP-AES or EDS. These tests can be costly and may not be readily available. Nevertheless, they are important to establish the material’s chemistry and mechanism of action. Without knowing the elemental composition of the studied CaP materials, it is impossible to establish a dose-effect relationship for the ion, which could result in ineffective or dangerously toxic materials. Regarding the latter possibility, testing cytotoxicity is essential to ensure its safety for patients. Without a measure of cytotoxicity, any reported antimicrobial effect is not a reliable indicator of a material’s suitability for clinical application.

Thirdly, the number of replicate experiments needs to be increased in most studies. There was large variability between studies in the number of replicates. While most studies performed three replicate experiments, one-third of the studies performed fewer than three. Three replicates should be considered a minimum number for future studies, with nine replicates being preferred (e.g., by including three replicates in three consecutive experiments). Experimental replicates validate the outcomes and establish the level of variance, allowing assessment of the statistical power of the results in the form of standard deviations or confidence intervals, enabling assessment of the clinical and statistical significance of the results [83].

### 4.3. Analysis of Quality Versus Study Age and Citations

No changes in reporting or methodological quality were observed over time, despite the length of the research period and the research output during that time. As there are still several areas of needed improvement in both reporting and methodological quality, there appears to be a need for guidelines on reporting and methodological quality of in vitro studies of antimicrobial biomaterials to improve quality in future research. Contrary to our expectation, reporting quality was not associated with citation rate. Due to the importance of adequate reporting quality for understanding and replicating research, an increasing trend in citation rate with higher reporting quality was expected. Conversely, higher methodological quality was associated with a higher citation rate, although only by a small effect.

### 4.4. Limitations & Strengths

This review has some limitations that should be considered. Firstly, it only includes studies published up to 6 December 2021. However, it is unlikely that literature published after 2021 would result in different outcomes, since no significant relation was found between study age and reporting quality or methodological quality. Secondly, some of the lowest-quality studies were already excluded from the dataset and not assessed, due to the inclusion and exclusion criteria for the dataset. Most notably, studies with non-quantitative or unintelligible data (reporting quality item 11), studies with error bars but no reporting of replicates (reporting quality item 14) and studies with composite materials or materials with phase impurities (reporting quality item 2 and methodological quality item 4) were excluded. Consequently, some of the options in our checklist represented 0 studies, because these studies were already excluded. The average quality found in this review is therefore likely to be higher than the true average.

Despite these limitations, this review presents the first quality assessment tool of reporting and methodological quality in biomaterials research. Although this checklist aimed to assess the reporting and methodological quality of studies reporting on antimicrobial calcium phosphate materials, these checklists could be adapted to studies on other types of antimicrobial biomaterial by adding or modifying quality items relevant to their respective fields. These adapted guidelines should cover minimum standards for characterisation and outline proper procedures for antimicrobial tests. In general, this checklist can improve the quality of future research in the field of antimicrobial CaP materials and therefore expedite the development of these materials for clinical use to help treat and prevent implant-related infections.

## 5. Conclusions

A checklist was created for assessing the reporting and methodological quality in research on antimicrobial CaP materials. This checklist was subsequently used to assess the quality of a dataset of existing studies in this field. There are several quality items that require improvement. For reporting quality, these are experimental reproducibility, reporting rationales for experiment design choices and discussing experimental results in relation to desired outcomes. For methodological quality, these are use of positive controls, increasing the number of replicate experiments and performing additional experiments on material chemistry and toxicity. The importance of the reporting and methodological quality checklist developed in our study is underscored by the lack of improvement in study quality over time. Using this checklist, future studies on antimicrobial CaPs can improve their quality, hopefully resulting in more in vivo studies and clinical trials on these materials.

## Figures and Tables

**Figure 1 materials-18-01543-f001:**
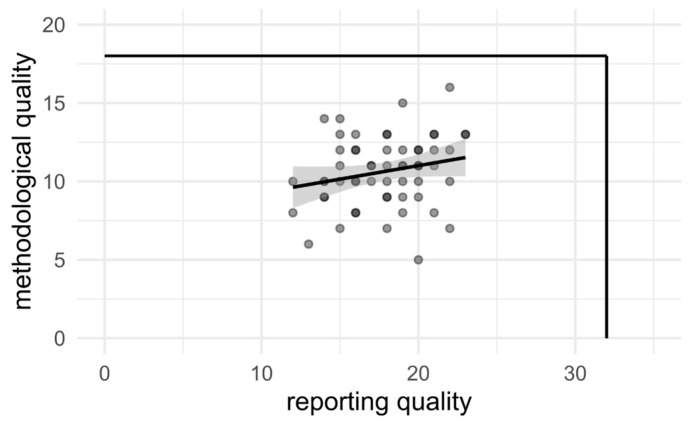
Methodological quality scores versus reporting quality scores of included studies. The black lines mark the maximum possible scores (18 points for methodological quality, 32 points for reporting quality).

**Figure 2 materials-18-01543-f002:**
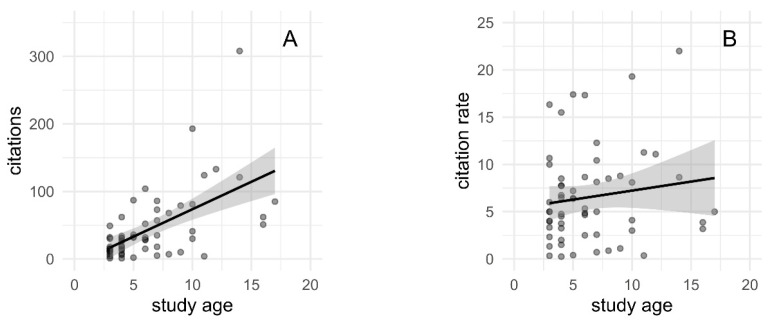
Study age versus citations (**A**) and citation rates (**B**).

**Figure 3 materials-18-01543-f003:**
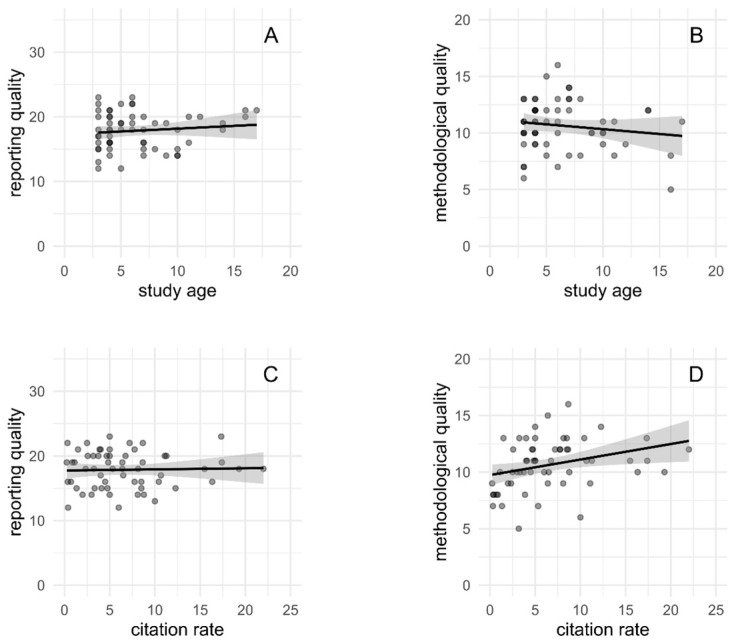
Reporting quality scores (**A**,**C**) and methodological quality scores (**B**,**D**) versus study age (**A**,**B**) and citation rate (**C**,**D**).

**Table 1 materials-18-01543-t001:** The reporting quality checklist.

**Reporting quality items**
**Use and reporting of positive and negative control groups**
1. How were negative control groups reported for the antimicrobial experiment?
Material characterisation
2. Was material phase purity (by XRD) adequately reported?
3. Were the results of other characterisation methods adequately reported?
**Reproducibility and validity of the antimicrobial tests**
4. Were all antimicrobial experiments and material characterisations that were described in the method also reported in the results?
5. Was there a study protocol?
6. Was the bacterial species reported?
7. Was the bacterial strain reported?
8. Was the bacterial challenge dose (starting inoculum) reported?
9. Was the amount of sample (e.g., mg/mL of powder or coating surface area) used reported in enough detail to reproduce the experiment?
10. Are other conditions used for antimicrobial tests reported in a reproducible manner?
**Reporting and discussion of the results of antimicrobial tests**
11. Were the results of antimicrobial tests reported in a readable manner?
12. Was bacterial enumeration reported on a logarithmic scale?
13. Was the clinical relevance of the observed antimicrobial effect discussed in relation to a desired outcome?
**Use of replicate experiments and statistics**
14. Was there clear reporting of the number of technical/biological replicates of the antimicrobial tests?
15. Was there a measure of variance adequately reported for of the antimicrobial tests?
16. Were the applied statistics for the antimicrobial tests appropriately described?
**Rationales for experiment design choices**
17. Was a clear clinical rationale provided for the chosen material formulation?
18. Was a rationale provided for the clinical relevance of the used microorganisms?
19. Was a clear rationale for the chosen antibacterial test method provided?
20. Was a bacterial killing mechanism proposed and substantiated?

**Table 2 materials-18-01543-t002:** The methodological quality checklist.

**Methodological quality items**
**Use and reporting of positive and negative control groups**
1. Was an appropriate negative control group used?
2. Was an appropriate positive control group used?
**Material characterisation**
3. What was the formulation of the antimicrobial material?
4. Was the material phase purity determined?
5. Were materials characterised using other methods as well?
6. Was the release of antimicrobial agents from the material investigated?
7. Did the authors quantify the elemental composition of the antimicrobial product?
**Reproducibility and validity of the antimicrobial tests**
8. Was the antimicrobial experiment performed on planktonic or surface-adherent bacteria?
9. Were multiple species of microorganisms used for the antimicrobial studies?
**Use of replicate experiments and statistics**
10. How many replicate experiments were performed?
**Testing of material toxicity**
11. How was the (non-)toxicity of the materials tested?

**Table 3 materials-18-01543-t003:** Reporting quality per item for the dataset of 58 studies.

Reporting Quality Item	Answers (Score)	Results N (%)
**Use and reporting of positive and negative control groups**		
1. How were negative control groups reported for the antimicrobial experiment?	A—Not reported/reported in method only (0) B—Results reported as relative versus negative control (1) C—Test samples and controls reported separately (2)	1 (2%) 4 (7%) 53 (91%)
**Material characterisation**		
2. Was material phase purity (by XRD) adequately reported?	A—No XRD results were reported (0) B—XRD was performed, but the pattern(s) not reported (0) C—XRD patterns were provided, but not discussed (1) D—the XRD patterns were reported and discussed (2)	7 (12%) 1 (2%) 0 (0%) 50 (86%)
3. Were the results of other characterisation methods adequately reported?	A—No other characterisation results were reported (0) B—Other characterisation was performed, but the results are not shown (0) C—The results of other characterisation were reported but not discussed (1) D—The results of other characterisations were reported and discussed (2)	2 (3%) 1 (2%) 0 (0%) 55 (95%)
**Reproducibility and validity of the antimicrobial tests**		
4. Were all antimicrobial experiments and material characterisations that were described in the method also reported in the results?	A—Not all performed experiments described in the method are reported in the results or vice versa (0) B—Some tests were only performed on a selection of the materials (1) C—All data are present and accounted for (2)	13 (22%) 25 (43%) 20 (34%)
5. Was there a study protocol?	A—No (0) B—Yes (1)	58 (100%) 0 (0%)
6. Was the bacterial species reported?	A—No (0) B—Yes (1)	0 (0%) 58 (100%)
7. Was the bacterial strain reported?	A—No (0) B—Yes (1)	25 (43%) 33 (57%)
8. Was the bacterial challenge dose (starting inoculum) reported?	A—No (0) B—Yes (1)	23 (40%) 35 (60)
9. Was the amount of sample (e.g., mg/mL of powder or coating surface area) used reported in enough detail to reproduce the experiment?	A—No (0) B—Yes (1)	22 (38%) 36 (62%)
10. Are other conditions used for antimicrobial tests reported in a reproducible manner?	A—Experimental conditions are not reported in a reproducible manner (0) B—Test were carried out according to a referenced procedure (1) C—The experiment is reported in sufficient detail that the experiment can be reproduced (2)	8 (14%) 7 (12%) 43 (74%)
**Reporting and discussion of the results of antimicrobial tests**		
11. Were the results of antimicrobial tests reported in a readable manner?	A—Data are unintelligible (e.g., 3D graphs) (0) B—Results are reported as images (1) C—Results reported quantitatively (tables, annotated graphs or written text) (2)	0 (0%) 37 (64%) 21 (36%)
12. Was bacterial enumeration reported on a logarithmic scale?	A—No bacteria count was performed (0) B—Enumeration was reported on a linear scale (1) C—Enumeration was performed on a logarithmic scale (2)	23 (40%) 22 (38%) 13 (22%)
13. Was the clinical relevance of the observed antimicrobial effect discussed in relation to a desired outcome?	A—No (0) B –Results were compared to those of other studies (1) C—The relevance of the antimicrobial results were discussed in relation to a desired outcome (2) D—The study reports that no consensus exists on the desired outcome (2)	49 (84%) 5 (9%) 3 (5%) 1 (2%)
**Use of replicate experiments and statistics**		
14. Was there clear reporting of the number of technical/biological replicates of the antimicrobial tests?	A—There are error bars, but the number of replicate experiments was not reported (0) B—There is no reporting of replicates (1) C—It is reported that there are replicates, but it is not clear if they are technical or biological replicates (1) D—The number of replicates is reported, and it is clear if they are technical or biological replicates (2)	0 (0%) 13 (22%) 9 (16%) 36 (62%)
15. Was there a measure of variance adequately reported for of the antimicrobial tests?	A—It is reported that there are replicate experiments, but only the mean outcome is reported (0) B—Variance is reported, but it is unclear what measure of uncertainty they represent (0) C—There is a measure of variance, and it is clear what it represents (1) D—There is no evidence of replicate experiments, and no measure of variance is provided (1) E—There are replicate measurements, and all raw data are available (2)	14 (24%) 11 (19%) 21 (36%) 12 (21%) 0 (0%)
16. Were the applied statistics for the antimicrobial tests appropriately described?	A—No statistics were applied, or results were not reported (0) B—Statistical values are used (e.g., *p*-values), but it is unclear by which method they were obtained (0) C—Statistical methods were applied, and it is clear which ones (1)	35 (60%) 4 (7%) 19 (33%)
**Rationales for experiment design choices**		
17. Was a clear clinical rationale provided for the chosen material formulation?	A—No (0) B—Yes (1)	45 (78%) 13 (22%)
18. Was a rationale provided for the clinical relevance of the used microorganisms?	A—No rationale was provided (0) B—A rationale was provided other than clinical (e.g., to cover Gram+ and Gram-) (1) C—A clinical rationale was provided (2)	22 (38%) 15 (26%) 21 (36%)
19. Was a clear rationale for the chosen antibacterial test method provided?	A—No (0) B—Yes (1)	54 (93%) 4 (7%)
20. Was a bacterial killing mechanism proposed and substantiated?	A—No killing mechanism was proposed (0) B—A theoretical killing mechanism was proposed but not substantiated (0) C—A theoretical killing mechanism was proposed based on references to literature (1) D—A killing mechanism was proposed and substantiated based on experimental evidence (2)	16 (28%) 8 (14%) 32 (55%) 2 (3%)

**Table 5 materials-18-01543-t005:** Results of linear regression models of reporting quality and methodological quality versus study age and citation rates.

	Reporting Quality		Methodological Quality	
	Slope	95% CI	Slope	95% CI
Study age	0.09	−0.11–0.29	−0.09	−0.24–0.07
Citation rate	0.02	−0.13–0.16	0.14	0.03–0.25

**Table 4 materials-18-01543-t004:** Methodological quality per item for the dataset of 58 studies.

Methodological Quality Item	Answers (Score)	Results N (%)
**Use and reporting of positive and negative control groups**		
1. Was an appropriate negative control group used?	A—No negative control was used/unclear negative control (0) B—A blank group was used as negative control (e.g., empty culture well) (1) C—A CaP material without antimicrobial effect was used as negative control (2) D—A different clinically relevant negative control was used (e.g., uncoated titanium or bone graft) (2)	3 (5%) 4 (7%) 49 (84%) 2 (3%)
2. Was an appropriate positive control group used?	A—No positive control was used (0) B—Antibiotics were used as a positive control group (1) C—A different positive control was used (1)	49 (84%) 7 (12%) 2 (3%)
**Material characterisation**		
3. What was the formulation of the antimicrobial material?	A—(Nano)powder (0) B—Coating (1) C—Pure CaP scaffold (1) D—Composite particles (1) E—Composite scaffold (1) F—Cement (1) G—Powders pressed into pellets or disks (0)	30 (52%) 13 (22%) 1 (2%) 0 (0%) 1 (2%) 1 (2%) 12 (21%)
4. Was the material phase purity determined?	A—No (0) B—Yes (1)	7 (12%) 51 (88%)
5. Were materials characterised using other methods as well?	A—No (0) B—Yes (1)	3 (5%) 55 (95%)
6. Was the release of antimicrobial agents from the material investigated?	A—No (0) B—Yes, by disk diffusion (1) C—Yes, by a specialised method (2)	27 (47%) 14 (24%) 17 (29%)
7. Did the authors quantify the elemental composition of the antimicrobial product?	A—There is no measure of the final product composition (0) B—The final composition is calculated based on the starting materials (0) C—The presence of the antimicrobial ion is confirmed (e.g., by XPS) but not quantified (1) D—An appropriate method has been used to measure the elemental composition, e.g., ICP-MS (2) E—The composition is determined indirectly, e.g., by Rietveld refinement (2)	6 (10%) 6 (10%) 10 (17%) 35 (60%) 1 (2%)
**Reproducibility and validity of the antimicrobial tests**		
8. Was the antimicrobial experiment performed on planktonic or surface-adherent bacteria?	A—Unclear if planktonic or adherent bacteria were measured (0) B—The antimicrobial assay was performed on planktonic bacteria (1) C—The antimicrobial assay was performed on surface-adherent bacteria (1) D—Both planktonic and surface-adherent bacteria were tested (2)	6 (10%) 42 (72%) 5 (9%) 5 (9%)
9. Were multiple species of microorganisms used for the antimicrobial studies?	A—Only 1 strain (0) B—Multiple strains with only 1 group (G−, G+ etc) (0) C—G+ and G− (1) D—G− or G+, and yeast (1) E—G+, and G−, and yeast (2)	9 (16%) 2 (3%) 36 (62%) 1 (2%) 10 (17%)
**Use of replicate experiments and statistics**		
10. How many replicate experiments were performed?	A—No replicates were performed or replicates were not reported (0) B—N = 2 (0) C—N ≥ 3 (1) D—N ≥ 9 (2)	16 (28%) 2 (3%) 40 (69%) 0 (0%)
**Testing of material toxicity**		
11. How was the (non-)toxicity of the materials tested?	A—No toxicity test was performed (0) B—Reference to a different study that rationalises why the current materials are or are not toxic (1) C—The study assessed a single measure of cell toxicity of the materials (1) D—The study assessed multiple of cellular/in vivo responses to the materials (2)	22 (38%) 3 (5%) 15 (26%) 18 (31%)

## Data Availability

The data that support the findings of this study (checklist, resulting scores) are openly available in the Harvard Dataverse at https://doi.org/10.7910/DVN/TYGLAU and https://doi.org/10.7910/DVN/KTD9EG.

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
