# Peer review of "Assessment of Quality in Antimicrobial Calcium Phosphate Research (AQUACAP): A Systematic Review"

_materials, 2025, doi:10.3390/ma18071543_

Round 1
Reviewer 1 Report
Comments and Suggestions for Authors
Introduction:
The introduction provides a strong foundation, but it would benefit from a more direct connection to clinical applications. Expanding on how the findings regarding antimicrobial CaP materials could influence clinical practices would help readers understand the broader relevance of the research.
Methods:
The checklist introduced in the Methods section is a valuable tool. However, the explanation of the scoring system could be more detailed. It would help if the authors clearly outlined the criteria used to assign scores, ensuring transparency and improving the reproducibility of the evaluation process.
Results:
The Results section is clear, but it could be enhanced by providing more explicit connections between the checklist items and the findings. Including examples of studies that scored highly or poorly would help readers better understand the variation in research quality.
Discussion:
The Discussion is informative, though it could benefit from a deeper analysis of the differences between reporting and methodological quality. The authors should provide concrete examples to illustrate how specific methodological flaws impacted the results. Additionally, a more thorough discussion of the clinical relevance of the findings would strengthen the connection between the experimental data and real-world applications.
Conclusion:
The Conclusion is brief but effective. However, it could be more impactful by emphasizing the broader significance of the study and how the checklist can guide future research. Mentioning potential next steps or future research directions would provide a forward-looking perspective to conclude the article.
The overall quality of the English in the article is solid; however, there are a few areas that could be refined to improve clarity and precision. Below are specific suggestions for improvement in each section:
Introduction:
The introduction is well-structured, but some phrases could be more concise. For example, the sentence “The checklist was developed to assess the reporting and methodological quality of antimicrobial CaP studies, and it is intended to be used as a tool for both assessing existing studies and guiding future research” could be simplified to: “The checklist was developed to assess the reporting and methodological quality of antimicrobial CaP studies and guide future research.” This revision would enhance clarity without losing meaning.
Methods:
While the Methods section is thorough, some technical terms could benefit from clearer explanations, especially for readers who may not be as familiar with them. For instance, the term “phase characterisation by powder X-ray diffraction (XRD)” could be clarified as follows: "X-ray diffraction (XRD), a technique used to determine the crystal structure of materials." This addition would help readers better understand the methodology.
Results:
In this section, the presentation is mostly clear, but some sentences could be more direct. For example, the sentence “This indicated that the methodological and reporting quality could be independently established by our checklist” could be rewritten as: “This indicates that the checklist can independently assess both methodological and reporting quality.” This adjustment would make the sentence more concise and straightforward.
Discussion:
The Discussion section is comprehensive; however, certain phrases can be streamlined for clarity. For instance, the sentence “Several included studies claimed ‘impressive’ or ‘significant’ outcomes without defining these terms” could be simplified to: “Several studies claimed ‘impressive’ or ‘significant’ outcomes but failed to define these terms.” This revision eliminates unnecessary words while maintaining the meaning.
In summary, the article demonstrates a strong command of English. By refining some of the language, particularly in the Methods and Discussion sections, the text would become more concise and accessible. Additionally, providing brief explanations for technical terms would improve readability and ensure that the content is clear to a broader audience.
Author Response
Dear reviewer,
Thank you for your time and effort in reading our manuscript and providing your comments. Based on your feedback, we have made several changes to the manuscript (outlined below).
Comment 1:
The introduction provides a strong foundation, but it would benefit from a more direct connection to clinical applications. Expanding on how the findings regarding antimicrobial CaP materials could influence clinical practices would help readers understand the broader relevance of the research.
Response 1:
Thank you for your suggestion. We have expanded the introduction section to include the current uses of CaP materials in the clinic as well as the outlook for future antimicrobial CaPs.
Comment 2:
The checklist introduced in the Methods section is a valuable tool. However, the explanation of the scoring system could be more detailed. It would help if the authors clearly outlined the criteria used to assign scores, ensuring transparency and improving the reproducibility of the evaluation process.
Response 2:
We have clarified our scoring process, and added examples for scoring in both reporting and methodological quality.
Comment 3:
The Results section is clear, but it could be enhanced by providing more explicit connections between the checklist items and the findings. Including examples of studies that scored highly or poorly would help readers better understand the variation in research quality.
Response 3:
Thank you for your suggestion. We have made several changes in our methods section to clarify the relationship between the overall study quality and individual quality items in the checklist. We would prefer not to provide explicit examples of studies with low quality, since our main aim is to improve quality in future studies. Additionally, citing a low-quality paper would mean we would direct traffic towards a study that provides a bad example for future research, based on our own data. Conversely, we have provided an example of a study with good reporting quality in our discussion section.
Comment 4:
The Discussion is informative, though it could benefit from a deeper analysis of the differences between reporting and methodological quality. The authors should provide concrete examples to illustrate how specific methodological flaws impacted the results. Additionally, a more thorough discussion of the clinical relevance of the findings would strengthen the connection between the experimental data and real-world applications.
Response 4:
Based on your suggestion, we have made some changes to the discussion section. Furthermore, we highlighted the clinical relevance of our study in the ‘limitations & strengths’ section of our manuscript.
Comment 5:
The Conclusion is brief but effective. However, it could be more impactful by emphasizing the broader significance of the study and how the checklist can guide future research. Mentioning potential next steps or future research directions would provide a forward-looking perspective to conclude the article.
Response 5:
Thank you for your suggestion, we have added to the future outlook section of our conclusion based on your feedback.
Comment 6:
The overall quality of the English in the article is solid; however, there are a few areas that could be refined to improve clarity and precision. Below are specific suggestions for improvement in each section:
Response 6:
We would like to thank you for your suggestions for improving the quality of our English. We have implemented the suggestions by you where possible, and amended the English in other places as well. However, we were confused by some of your suggestions. Specifically, suggestion 1 and 3 are in different sections of the manuscript than your comments indicate. Secondly, we have already included a brief explanation on the purpose of XRD analysis in the discussion (suggestion 2). Nevertheless, we have made changes to the referenced sections.
Thank you again for your feedback. We hope we have satisfied your with our responses and the changes made to the manuscript.
Reviewer 2 Report
Comments and Suggestions for Authors
The Authors describe a novel checklist to identify key areas for improvement of reporting and methodological quality in the field of ion-substituted antimicrobial calcium phosphates.
The topic is interesting and the study well designed.
How was the checklist developed? Please detail further.
Would include all studies in the analysis.
Author Response
Dear reviewer,
Thank you for your time and effort in reading our manuscript and providing your comments. Based on your feedback, we have made some changes to the manuscript (outlined below).
Comment 1:
How was the checklist developed? Please detail further.
Response 1:
Thank you for your suggestion. Based on your feedback, we have expanded the ‘design of the checklist’ section to include more details on the creation process for the checklist.
Comment 2:
Would include all studies in the analysis.
Response 2:
We have decided to include 58 studies in our review, based on the need to strike a balance between time required to gather the data, and the amount of data necessary to find trends in methodological and reporting quality. We are reluctant to expand our database due to the associated workload while we expect the return on the investment to be minor. As we have shown in our manuscript, there are no observable trends in study quality over time and the included studies were randomized; therefore, we do not expect that expanding the database would change the main outcomes of the study..
Thank you again for your feedback. We hope we have satisfied your with our responses and the changes made to the manuscript.
Reviewer 3 Report
Comments and Suggestions for Authors
The manuscript presents a well-structured and relevant systematic review assessing the reporting and methodological quality of studies on antimicrobial ion-substituted calcium phosphate biomaterials. The introduction effectively contextualizes the need for a standardized quality checklist and clearly outlines the research objectives. The methodology is detailed and transparent, adhering to PRISMA guidelines, which strengthens the study’s reproducibility. The results provide a comprehensive analysis of reporting and methodological quality in the field, identifying key areas for improvement. The discussion effectively interprets the findings and highlights the importance of standardized reporting and study design in biomaterial research.
Despite its strengths, a few areas require minor revisions for clarity and completeness. In the abstract, while the key findings are well summarized, including a brief mention of the implications for future research would enhance its impact. The methodology section could benefit from a clearer explanation of the scoring system used for the checklist, particularly how different answer choices contribute to the final score. Additionally, specifying the criteria used to distinguish between high- and low-quality studies would improve transparency.
In the results section, while the quartile-based comparison is useful, a clearer presentation of key numerical differences in reporting and methodological quality would improve readability. A table summarizing the most common deficiencies across studies would help highlight critical areas for improvement. Furthermore, a brief explanation of why certain methodological aspects, such as the use of positive control groups, scored consistently low would strengthen the analysis.
The discussion is well-developed but could benefit from a more explicit statement on how the proposed checklist compares to existing quality assessment tools in biomaterial research. Additionally, while the manuscript mentions the lack of improvement in study quality over time, briefly discussing potential reasons for this trend and how the checklist could address these challenges would enhance the manuscript’s impact.
The conclusion effectively summarizes the study’s main contributions but could be slightly expanded to emphasize the broader implications for improving research quality in antimicrobial biomaterials. Adding a brief statement on how this checklist could be adapted for other biomaterial studies would further strengthen its significance.
Author Response
Dear reviewer,
Thank you for your time and effort in reading our manuscript and providing your comments. Based on your feedback, we have made several changes to the manuscript (outlined below).
Comment 1:
Despite its strengths, a few areas require minor revisions for clarity and completeness. In the abstract, while the key findings are well summarized, including a brief mention of the implications for future research would enhance its impact.
Response 1:
Thank you for your suggestion, we have made some changes to the abstract based on your feedback.
Comment 2:
The methodology section could benefit from a clearer explanation of the scoring system used for the checklist, particularly how different answer choices contribute to the final score. Additionally, specifying the criteria used to distinguish between high- and low-quality studies would improve transparency.
Response 2:
Based on your feedback, we have improved the ‘design of the checklist’ section to better reflect the design process and the scoring system, as well as the way we distinguish high- and low-scoring studies.
Comment 3:
In the results section, while the quartile-based comparison is useful, a clearer presentation of key numerical differences in reporting and methodological quality would improve readability. A table summarizing the most common deficiencies across studies would help highlight critical areas for improvement. Furthermore, a brief explanation of why certain methodological aspects, such as the use of positive control groups, scored consistently low would strengthen the analysis.
Response 3:
Thank you for your comment. We would like to avoid speculating on the reasons why studies score high or low on certain quality items, but we have incorporated some possibilities in our manuscript.
Comment 4:
The discussion is well-developed but could benefit from a more explicit statement on how the proposed checklist compares to existing quality assessment tools in biomaterial research. Additionally, while the manuscript mentions the lack of improvement in study quality over time, briefly discussing potential reasons for this trend and how the checklist could address these challenges would enhance the manuscript’s impact.
Response 4:
Unfortunately, there are no existing tools for assessing quality in biomaterials research. This dearth of quality assessment tools is what led to the creation of our manuscript. We have clarified this in the ‘limitations & strengths’ section of our manuscript.
Comment 5:
The conclusion effectively summarizes the study’s main contributions but could be slightly expanded to emphasize the broader implications for improving research quality in antimicrobial biomaterials. Adding a brief statement on how this checklist could be adapted for other biomaterial studies would further strengthen its significance.
Response 5:
Thank you for these excellent suggestions. We have made changes to our manuscript to incorporate your feedback in the ‘conclusions’ and ‘limitations & strengths’ sections of our manuscript.
Thank you again for your feedback. We hope we have satisfied your with our responses and the changes made to the manuscript.
Reviewer 4 Report
Comments and Suggestions for Authors
The article aims to develop a novel checklist to assess the quality of preclinical studies on antimicrobial calcium phosphates with substituted ions. It will also use this checklist to assess the quality of the existing literature and identify areas for improvement. The checklist focuses on two main aspects: the quality of the reports and the methodological quality.
The article is interesting and well-developed from the point of view of communication of scientific activity.
My main doubt is the selection and identification of the panel of experts. This section should be written in greater depth (the experts could be identified, and if that were not possible, at least their institutions or countries of origin). In the same way, all of them should be included in the list of acknowledgments to the article.
Author Response
Dear reviewer,
Thank you for your time and effort in reading our manuscript and providing your comments. Based on your feedback, we have made several changes to the manuscript (outlined below).
Comment 1:
My main doubt is the selection and identification of the panel of experts. This section should be written in greater depth (the experts could be identified, and if that were not possible, at least their institutions or countries of origin). In the same way, all of them should be included in the list of acknowledgments to the article.
Response 1:
thank you for your suggestion. All experts that were involved in this study are either co-authors, or are acknowledged as such. We have found that there was a mistake in the formatting of the ‘acknowledgements’ section, which could have led to confusion for the reader. We have resolved the issue, and also modified the ‘design of the checklist’ section to better clarify the identity of the experts.
Thank you again for your feedback. We hope we have satisfied your with our responses and the changes made to the manuscript.